# Logical Activation Functions: Logit-space equivalents of Probabilistic Boolean Operators

**Scott C. Lowe**[1,2,*]**, Robert Earle**[1,2]**, Jason d'Eon**[1,2]**, Thomas Trappenberg**[1]**, Sageev Oore**[1,2]

[1]Faculty of Computer Science
Dalhousie University
Halifax, Nova Scotia
Canada

[2]Vector Institute for Artificial Intelligence
Toronto, Ontario
Canada

[*]Correspondence: scottclowe@gmail.com

## Abstract

The choice of activation functions and their motivation is a long-standing issue within the neural network community. Neuronal representations within artificial neural networks are commonly understood as logits, representing the log-odds score of presence of features within the stimulus. We derive logit-space operators equivalent to probabilistic Boolean logic-gates AND, OR, and XNOR for independent probabilities. Such theories are important to formalize more complex dendritic operations in real neurons, and these operations can be used as activation functions within a neural network, introducing probabilistic Boolean-logic as the core operation of the neural network. Since these functions involve taking multiple exponents and logarithms, they are computationally expensive and not well suited to be directly used within neural networks. Consequently, we construct efficient approximations named $\text{AND}_\text{AIL}$ (the AND operator Approximate for Independent Logits), $\text{OR}_\text{AIL}$, and $\text{XNOR}_\text{AIL}$, which utilize only comparison and addition operations, have well-behaved gradients, and can be deployed as activation functions in neural networks. Like MaxOut, $\text{AND}_\text{AIL}$ and $\text{OR}_\text{AIL}$ are generalizations of ReLU to two-dimensions. While our primary aim is to formalize dendritic computations within a logit-space probabilistic-Boolean framework, we deploy these new activation functions, both in isolation and in conjunction to demonstrate their effectiveness on a variety of tasks including tabular classification, image classification, transfer learning, abstract reasoning, and compositional zero-shot learning.

## 1 Introduction

Non-linear activation functions are essential in artificial neural networks (ANNs) to form higher-order representations, since otherwise the network would be degeneratively equivalent to a single linear layer. Most activation functions represent a simple non-linearity despite evidence of much more complex non-linear integration and computations in dendrites (Hentschel et al., 2004; London & Häusser, 2005; Payeur et al., 2019). For example, Gidon et al. (2020) recently demonstrated that a single biological neuron can compute the XOR of its inputs, a property long known to be lacking from artificial neurons (Minsky & Papert, 1969). While simple activation functions work in ANNs in the sense that more complex operations can be formed from the combination of several layers of neurons, understanding the function and impact of advanced dendritic operations in networks is important. In this work, we add some of this behaviour to neural activations, corresponding to shifting

36th Conference on Neural Information Processing Systems (NeurIPS 2022).

some of the network's complexity from its global structure to the neural level. There is far more complexity in biological neurons than in the abstractions that we consider here, but we make a step in the direction of using more complex neurons in ANNs. We also develop a theoretical underpinning for higher-order activation functions (e.g. like MaxOut, Goodfellow et al., 2013) in a probabilistic framework. We hypothesize that such architectures will be more parameter-efficient in situations where their assumptions hold.

Neuronal representations within ANNs are commonly understood as logits, representing the log-odds score of presence (versus absence) of features within the stimulus. From a Bayesian perspective, a ReLU-like operation corresponds to the removal of all evidence for the lack of a feature. Under the logit interpretation of ANN potentiations, this seems unreasonable. This can be seen in some of the ways we interact with neural networks: we must apply batch-norm before activations and not after them; when doing transfer learning from an embedding space we must use pre-activation potentiations instead of activations. Can ANNs do better if we design an architecture which treats potentiations as logits?

We thus derive logit-space operators equivalent to probabilistic Boolean logic-gates AND, OR, and XNOR for independent probabilities. Networks constructed in this way can be interpreted as **performing logical operations** using **point-estimates of probabilities**, in a similar manner to a Bayesian network. This brings operations from the symbolism framework of AI (which is more similar to deliberative thinking, or System 2 of Kahneman, 2011) into the connectionist framework of ANNs (which is more like instinctive, System 1, thinking). We also construct computationally feasible approximations to these functions with well-behaved gradients. These new activation functions, which are generalizations of ReLU to two-dimensions, are then applied on benchmark datasets to demonstrate their effectiveness on a diverse range of tasks including image classification, transfer learning, abstract reasoning, and compositional zero-shot learning. The new principled approach we present introduces new ways to redistribute computation from the network into the neuronal mechanisms, and build more parameter-efficient models. We demonstrate the effectiveness of activation functions based on these ideas, and expect future work to build on this.

## 2 Background

Early artificial neural networks featured either logistic-sigmoid or tanh as their activation function, motivated by the idea that each layer of the network is building another layer of abstraction of the stimulus space from the last layer. Each neuron in a layer identifies whether certain properties or features are present within the stimulus, and the pre-activation (potentiation) value of the neuron indicates a score or logit for the presence of that feature. The sigmoid function, $\sigma(x) = 1/(1+e^{-x})$, was hence a natural choice of activation function, since as with logistic regression, this will convert the logits of features into probabilities. There is evidence that this interpretation has merit. Previous work has been done to identify which features neurons are tuned to. Examples include LSTM neurons tracking quotation marks, line length, and brackets (Karpathy et al., 2015) and sentiment (Radford et al., 2017); projecting features back to the input space to view them (Olah et al., 2017); and interpretable combinations of neural activities (Olah et al., 2020).

Sigmoidal activation functions are no longer commonly used within machine learning between layers of representations, though sigmoid is still widely used for gating operations which scale the magnitude of other features in an attention-like manner. The primary disadvantage of the sigmoid activation function is its vanishing gradient — as the potentiation rises, activity converges to a plateau, and hence the gradient goes to zero. This prevents feedback information propagating back through the network from the loss function to the early layers of the network, which consequently prevents it from learning to complete the task.

The Rectified Linear Unit activation function (Fukushima, 1980; Jarrett et al., 2009; Nair & Hinton, 2010), $\mathrm{ReLU}(x) = \max(0, x)$, does not have this problem, since in its non-zero regime it has a gradient of 1. Another advantage of ReLU is it has very low computational demands. Since it is both effective and efficient, it has proven to be a highly popular choice of activation function. The chief drawback to ReLU is that it has no sensitivity to changes across half of its input domain, which prevents updates on stimuli which trigger its "off" state and can even lead to neuronal death[1]. Variants of ReLU have emerged, aiming to smooth out its transition between domains and provide

---

[1]Though this problem is very rare when using BatchNorm to stabilize feature distributions.

a gradient in its inactive regime. These include ELU (Clevert et al., 2016), CELU (Barron, 2017), SELU (Klambauer et al., 2017), GELU (Hendrycks & Gimpel, 2020), SiLU (Elfwing et al., 2018; Ramachandran et al., 2017), and Mish (Misra, 2019). However, all these activation functions still bear the general shape of ReLU and truncate negative logits.

Fuzzy logic operators are generalizations of Boolean logic operations to continuous variables, using rules similar to applying logical operators in probability space. Prior work has explored networks of fuzzy logic operations, including some which use an activation function constituting a learnable interpolation between fuzzy logic operators (Godfrey & Gashler, 2017). The activation functions we introduce here are motivated similarly to fuzzy logic operators, but designed to operate in logit space instead of in probability space, which better reflects the behaviour and space of pre-activation units.

Bayesian neural networks (BNNs) are probabilistic models which can represent the uncertainty in their model parameters, and hence uncertainty in their outputs. This differs from standard ANNs which use only a single point-estimate of each model parameter, and hence also in their neural activations. By making their priors explicit and modelling their uncertainty, Bayesian networks can be better calibrated and less vulnerable to overfitting than ANNs. However, BNNs are more challenging to train, and cannot reasonably be scaled up to the deep architectures which are possible with standard ANNs and necessary in order to learn a sufficiently complex model to solve highly complex tasks.

In this work we contribute to the theoretical underpinning of neural activation functions by developing activation functions based on the principle that neurons encode logits — scores that represent the presence of features in the log-odds space. In §3 we derive and define these functions in detail for different logical operators, and then consider their performance on numerous task types including parity (§4.1), image classification (§4.3 and §4.4), transfer learning (§4.5), abstract reasoning (Appendix §A.18), soft-rule guided classification as exemplified by the Bach chorale dataset (§4.2), and compositional zero-shot learning (Appendix §A.19). These tasks were selected to (1) survey the performance of the new activations on existing benchmark tasks, and (2) evaluate their performance on tasks which we suspect in particular may require logical reasoning and hence benefit from activation functions which apply these logical operations to logits.

## 3    Derivation

Manipulation of probabilities in logit-space is known to be more efficient for many calculations. For instance, the log-odds form of Bayes' Rule (Appendix Eq. 9) states that the posterior logit equals the prior logit plus the log of the likelihood ratio for the new evidence (the log of the Bayes factor). Thus, working in logit-space allows us to perform Bayesian updates on many sources of evidence simultaneously, merely by summing together the log-likelihood ratios for the evidence (see Appendix §A.3). A weighted sum may be used if the amount of credence given to the sources differs — and this is precisely the operation performed by a linear layer in a neural network.

When considering sets of probabilities, a natural operation is to measure the joint probability of two events both occurring— the AND operation. Suppose our input space is $x \in [0, 1]^2$, and the goal is to output $y > 0$ if $x_i = 1 \, \forall \, i$, and $y < 0$ otherwise, using model with a weight vector $w$ and bias term $b$, such that $y = w^T x + b$. This can be trivially solved with the weight matrix $w = [1, 1]$ and bias term $b = -1.5$. However, since this is only a linear separator, the solution can not generalize to the case $y > 0$ iff $x_i > 0 \, \forall \, i$. Similarly, let us consider how the OR function is solved with a linear layer. Our goal is to output $y > 0$ if $\exists \, x_i = 1$, and $y < 0$ otherwise. The binary case can be trivially solved with the weight matrix $w = [1, 1]$ and bias term $b = -0.5$. The difference between the solution for OR and the solution for AND is only an offset to our bias term. In each case, if the input space is expanded beyond binary to $\mathbb{R}^2$, the output can be violated by changing only one of the arguments.

### 3.1    AND

Suppose we are given $x$ and $y$ as the logits for the presence (vs absence) of two events, $X$ and $Y$. These logits have equivalent probability values, which can be obtained using the sigmoid function, $\sigma(u) = (1 + e^{-u})^{-1}$. Let us assume that the events $X$ and $Y$ are independent of each other. In this case, the probability of both events occurring (the joint probability) is $P(X, Y) = P(X \wedge Y) = P(X) P(Y) = \sigma(x) \sigma(y)$. However, we wish to remain in logit-space, and must determine the logit

of the joint probability, $\mathrm{logit}(\mathrm{P}(X,Y))$. This is given by

$$\mathrm{AND_{IL}} := \mathrm{logit}(\mathrm{P}(X \wedge Y)_{x \perp\!\!\!\perp y}) = \log\left(\frac{p}{1-p}\right), \text{ where } p = \sigma(x)\,\sigma(y),$$

$$= \log\left(\frac{\sigma(x)\,\sigma(y)}{1 - \sigma(x)\,\sigma(y)}\right), \tag{1}$$

which we coin as $\mathrm{AND_{IL}}$, the AND operator for independent logits (IL). This 2d function is illustrated as a contour plot (Fig. 1, left panel). Across the plane, the order of magnitude of the output is the same as at least one of the two inputs, scaling approximately linearly.

The approximately linear behaviour of the function is suitable for use as an activation function (no vanishing gradient), however taking exponents and logs scales poorly from a computational perspective. Hence, we developed a computationally efficient approximation as follows. Observe that we can loosely approximate $\mathrm{AND_{IL}}$ with the minimum function (Fig. 1, right panel). This is equivalent to assuming the probability of **both** $X$ and $Y$ being true equals the probability of the **least likely** of $X$ and $Y$ being true — a naïve approximation which holds well in three quadrants of the plane, but overestimates the probability when both $X$ and $Y$ are unlikely. In this quadrant, when both $X$ and $Y$ are both unlikely, a better approximation for $\mathrm{AND_{IL}}$ is the sum of their logits.

We thus propose $\mathrm{AND_{AIL}}$, a linear-approximate AND function for independent logits (AIL, i.e. approximate IL).

$$\mathrm{AND_{AIL}}(x,y) := \begin{cases} x + y, & x < 0,\, y < 0 \\ \min(x,y), & \text{otherwise} \end{cases} \tag{2}$$

As shown in Fig. 1 (left, middle), we observe that their output values and shape are very similar.

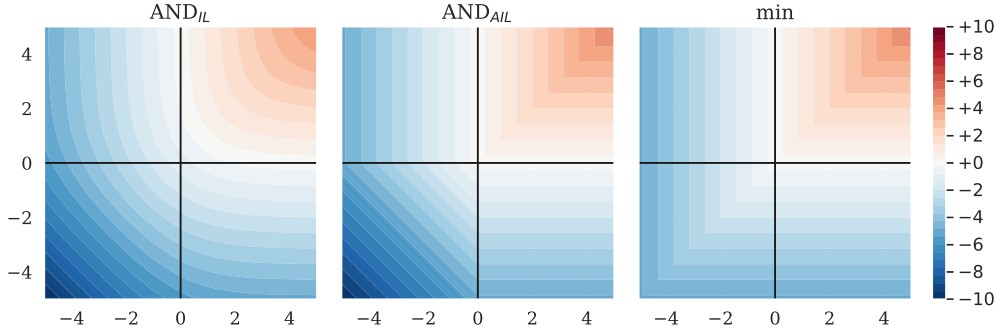

Figure 1: Heatmap comparing the outputs for the exact logit-space probabilistic-and function for independent logits, $\mathrm{AND_{IL}}(x,y)$; our constructed approximation, $\mathrm{AND_{AIL}}(x,y)$; and $\max(x,y)$.

## 3.2 OR

Similarly, we can construct the logit-space OR function, for independent logits. For a pair of logits $x$ and $y$, the probability that either of the corresponding events is true is given by $p = 1 - \sigma(-x)\,\sigma(-y)$. This can be converted into a logit as

$$\mathrm{OR_{IL}}(x,y) := \mathrm{logit}(\mathrm{P}(X \vee Y)_{x \perp\!\!\!\perp y}) = \log\left(\frac{p}{1-p}\right), \text{ with } p = 1 - \sigma(-x)\,\sigma(-y) \tag{3}$$

which can be roughly approximated by the max function. This is equivalent to setting the probability of **either** of event $X$ or $Y$ occurring to be equal to the probability of the **most likely** event. This underestimates the upper-right quadrant (below), which we can approximate better as the sum of the input logits, yielding

$$\mathrm{OR_{AIL}}(x,y) := \begin{cases} x + y, & x > 0,\, y > 0 \\ \max(x,y), & \text{otherwise} \end{cases} \tag{4}$$

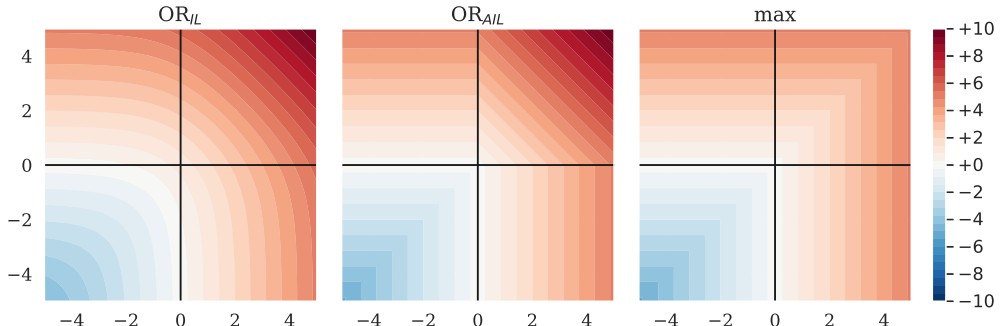

Figure 2: Comparison of the exact logit-space probabilistic-or function for independent logits, $\text{OR}_{\text{IL}}(x, y)$; our constructed approximation, $\text{OR}_{\text{AIL}}(x, y)$; and $\max(x, y)$.

### 3.3 XNOR

We also consider the construction of a logit-space XNOR operator. This is the probability that $X$ and $Y$ occur either together, or not at all, given by

$$\text{XNOR}_{\text{IL}}(x, y) := \text{logit}(\text{P}(X \overline{\oplus} Y)_{x \perp y}) = \log\left(\frac{p}{1-p}\right), \tag{5}$$

where $p = \sigma(x)\,\sigma(y) + \sigma(-x)\,\sigma(-y)$. We can approximate this with

$$\text{XNOR}_{\text{AIL}}(x, y) := \text{sgn}(xy)\min(|x|, |y|), \tag{6}$$

which focuses on the logit of the feature **most likely** to **flip** the expected **parity** (see Fig. 3).

We could use other approximations, such as the sign-preserving geometric mean,

$$\text{SignedGeomean}(x, y) := \text{sgn}(xy)\sqrt{|xy|}, \tag{7}$$

but this matches $\text{XNOR}_{\text{IL}}$ less closely, and has a divergent gradient along both $x = 0$ and $y = 0$.

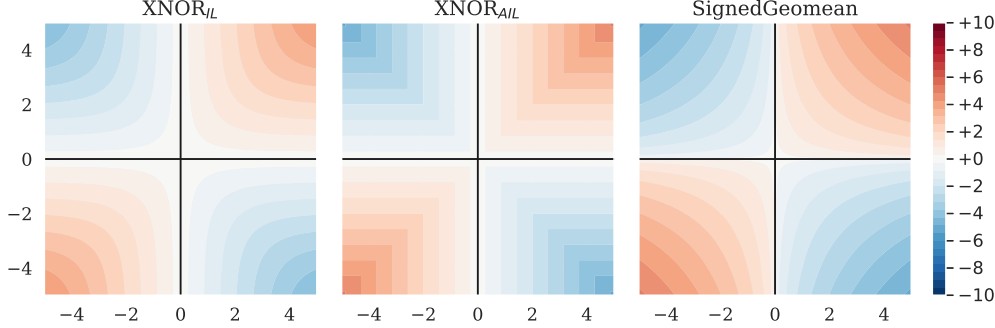

Figure 3: Comparison of the exact logit-space probabilistic-xnor function for independent logits, $\text{XNOR}_{\text{IL}}(x, y)$; our constructed approximation, $\text{XNOR}_{\text{AIL}}(x, y)$; and $\text{SignedGeomean}(x, y)$.

### 3.4 Discussion

By working via probabilities, and assuming inputs encode independent events, we have derived logit-space equivalents of the Boolean logic functions, AND, OR, and XNOR. Since these are computationally demanding to compute repeatedly within a neural network, we have constructed approximations of them: $\text{AND}_{\text{AIL}}$, $\text{OR}_{\text{AIL}}$, and $\text{XNOR}_{\text{AIL}}$. Like ReLU, these involve only comparison, addition, and multiplication operations which are cheap to perform. In fact, $\text{AND}_{\text{AIL}}$ and $\text{OR}_{\text{AIL}}$ are a generalization of ReLU to an extra dimension, since $\text{OR}_{\text{AIL}}(x, y = 0) = max(x, 0)$.

The majority of activation functions are one-dimensional, $f : \mathbb{R} \to \mathbb{R}$. In contrast to this, our proposed activation functions are all two-dimensional, $f : \mathbb{R}^2 \to \mathbb{R}$. They must be applied to pairs of features

from the embedding space, and will reduce the dimensionality of the input space by a factor of 2. This behaviour is the same as seen in MaxOut networks (Goodfellow et al., 2013) which use $\max$ as their activation function; $\text{MaxOut}(x, y; k) := \max(x, y)$. Similar to $\text{MaxOut}$, our activation functions could be generalized to higher dimensional inputs, $f : \mathbb{R}^k \to \mathbb{R}$, by considering the behaviour of the logit-space AND, OR, XNOR operations with regard to more inputs. For simplicity, we restrict this work to consider only $k\!=\!2$, but note these activation functions also generalize to higher dimensions.

### 3.5 Ensembling

By using multiple logit-Boolean activation functions *simultaneously* alongside each other, we permit the network multiple options of how to relate features together. When combining the activation functions, we considered two strategies (illustrated in Appendix §A.6).

In the partition (p) strategy, we split the $n_c$ dimensional pre-activation embedding equally into $m$ partitions, apply different activation functions on each partition, and concatenate the results together. Using AIL activation functions under this strategy, the output dimension will always be half that of the input, as it is for each AIL activation function individually. In the duplication (d) strategy, we apply $m$ different activation functions in parallel to the same $n_c$ elements. The output is consequently larger, with dimension $m\,n_c$. If desired, we can counteract the $2 \to 1$ reduction of AIL activation functions by using two of them together under this strategy. A negative weight in a network is equivalent to the logit-NOT operator. Hence with sufficient width and depth, a multi-layer network using only the $\text{OR}_{\text{IL}}$ activation function should be able to replicate any probabilistic logic circuit.

Utilizing $\text{AND}_{\text{AIL}}$, $\text{OR}_{\text{AIL}}$ and $\text{XNOR}_{\text{AIL}}$ simultaneously allows our networks to access logit-space equivalents of 12 of the 16 Boolean logical operations with only a single sign inversion (in either one of the inputs or the output). Including the bias term and skip connections, the network has easy access to logit-space equivalents of all 16 Boolean logical operations.

We hypothesised that training a network with all three of our activation functions in an ensemble in this manner could yield better results since the network would not need to expend layers having to combine $\text{OR}_{\text{IL}}$ operations together to yield other Boolean operations.

### 3.6 Normalization

Our AIL activation functions are close approximations to exact AND, OR, and XNOR operations in logit-space. However, when deploying non-linearities within a neural network, it is important that the activation functions have a gain of 1 in order to improve stability during training (Klambauer et al., 2017), a property the AIL activations do not possess. We constructed normalized variants of the exact and approximate logit operators, dubbed NIL and NAIL respectively, by subtracting the expected mean and dividing by the expected standard deviation, assuming the operands are sampled independently from the standard normal distribution, $\mathcal{N}(0, 1)$. For more details, see Appendix §A.7.

## 4 Experiments

We evaluated the performance of our AIL activations, both individually and together in an ensemble, on a range of benchmark tasks. Since $\text{AND}_{\text{AIL}}$ and $\text{OR}_{\text{AIL}}$ are equivalent when the sign of operands and outputs can be freely chosen, we only show results for $\text{OR}_{\text{AIL}}$. We compared against three primary baselines: (1) ReLU, (2) $\max(x, y) = \text{MaxOut}([x, y]; k\!=\!2)$, and (3) the concatenation of $\max(x, y)$ and $\min(x, y)$, denoted $\{\text{Max}, \text{Min (d)}\}$. The $\{\text{Max}, \text{Min (d)}\}$ ensemble is equivalent to $\text{GroupSort}$ with a group size of 2 (Chernodub & Nowicki, 2017; Anil et al., 2019), sometimes referred to as the $\text{MaxMin}$ operator; it is comparable to the concatenation of $\text{OR}_{\text{AIL}}$ and $\text{AND}_{\text{AIL}}$ under our duplication strategy.

Source code for our experiments can be found at https://github.com/DalhousieAI/logical_activation_experiments. Our python package which provides an implementation of these the activation functions is available at https://github.com/DalhousieAI/pytorch-logit-logic, which is also available as PyPI package pytorch-logit-logic.

## 4.1 Parity

In a simple initial experiment, we constructed a synthetic dataset whose labels could be derived directly by stacking the logical operation XNOR. Each sample had four input logits, and target value equal to the parity of the number of positive inputs. A very small model using XNOR with two hidden layers (of 4, then 2 neurons) should be capable of perfect classification accuracy on this dataset with a sparse weight matrix by learning to nest pairwise binary relationships. We trained such an MLP with either $\mathrm{ReLU}$ or $\mathrm{XNOR_{AIL}}$ activations.

The MLP with $\mathrm{XNOR_{AIL}}$ activation learned a sparse weight matrix able to perfectly classify any input combination, shown in Fig. 4. In comparison, with $\mathrm{ReLU}$ the network was only able to produce 60% classification accuracy. The accuracy with $\mathrm{ReLU}$ was improved by increasing the MLP width/depth, but this did not result in a sparse weight matrix. This experiment demonstrates that $\mathrm{XNOR_{AIL}}$ can be utilized by a network to find the simplest relationship between inputs that satisfies the objective. For additional results, see Appendix §A.11.

## 4.2 MLP on Bach Chorales and Logit Independence

The Bach Chorale dataset (Boulanger-Lewandowski et al., 2012) consists of 382 chorales composed by JS Bach, each ~12 measures long, totalling ~83 000 notes. Represented as discrete sequences of tokens, it has served as a benchmark for music processing for decades, from heuristic methods to HMMs, RNNs, and CNNs (Mozer, 1990; Hild et al., 1991; Allan & Williams, 2005; Liang, 2016; Hadjeres et al., 2017; Huang et al., 2019). The chorales are comprised of 4 voices (melodic lines) whose behaviour is guided by soft musical rules depending on the prior movement of that and other voices, e.g. "two voices a fifth apart ought not to move in parallel with one another". We tasked 2-layer MLPs with determining whether a short four-part musical excerpt is taken from a Bach chorale. Negative examples were created by stochastically corrupting chorale excerpts (see Appendix §A.13). We found $\{\mathrm{OR}, \mathrm{AND}, \mathrm{XNOR_{AIL}}$ (d)$\}$ had highest accuracy, but the results were not statistically significant ($p < 0.1$ between best and worst, two-tailed Student's $t$-test, 10 random inits; see Appendix §A.10).

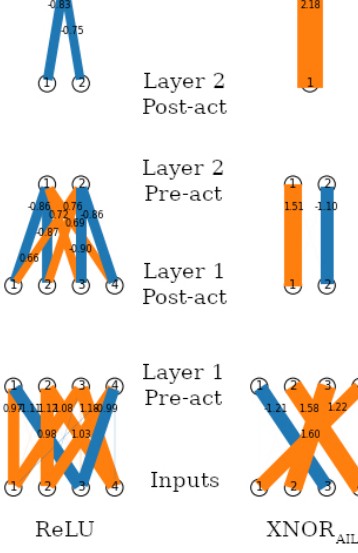

Figure 4: Visualization of weight matrices learnt by two-layer MLPs on a binary classification task, where the target output is the parity of the inputs. Line widths indicate weight magnitudes (orange: +ve, blue: -ve). MLP with ReLU: 60% accuracy; $\mathrm{XNOR_{AIL}}$: 100% accuracy.

Additionally, we investigated the independence between logits in the trained pre-activation embeddings. We expect that an MLP which is optimally engaging its neurons would maintain independence between features in order to maximize information. We measured correlations between adjacent pre-activations (paired operands for the logical activation functions), and also between non-adjacent pairs of pre-activations. Our results indicate the network learns features which are independent when they are not passed to the same 2D activation, and *anti-correlated* features when they are. For more details, see Appendix §A.14.

## 4.3 CNN and MLP on MNIST

We trained 2-layer MLP and 6-layer CNN models on MNIST with ADAM (Kingma & Ba, 2015), 1-cycle schedule (Smith & Topin, 2017; Smith, 2018), and using hyperparameters tuned through a random search against a validation set comprised of the last 10k images of the training partition.

The MLP used two hidden layers, the widths of which were varied together to evaluate the performance for a range of model sizes. The CNN used six layers of 3x3 convolution layers, with 2x2 max pooling (stride 2) after every other conv layer, followed by flattening and three MLP layers. The layer widths were scaled up to explore a range of model sizes (see Appendix §A.15 for more details).

For the MLP, $\mathrm{XNOR_{AIL}}$ performed best along with $\mathrm{SignedGeomean}$ ($p < 0.1$, two-tailed Student's $t$-test), ahead of all other activations ($p < 0.01$ for each; Fig. 5 left panel) when considering the best

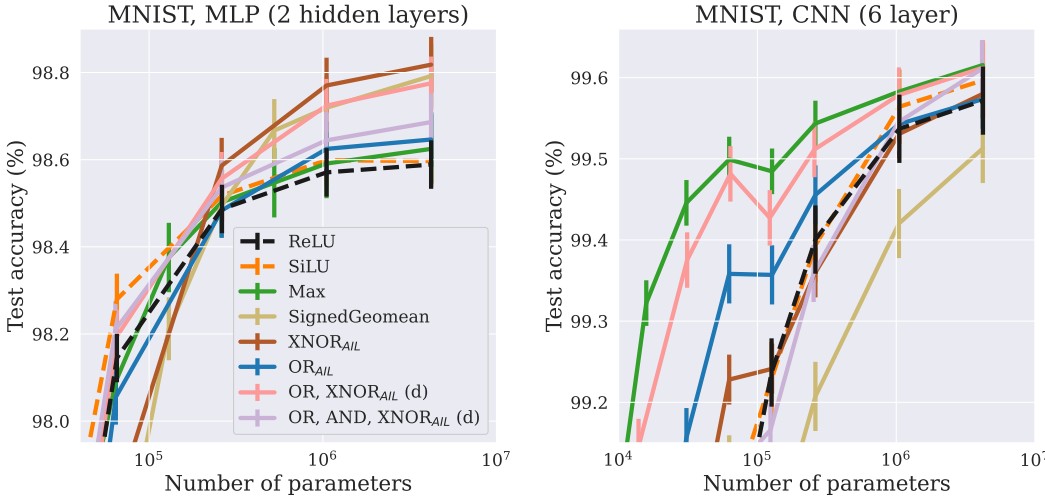

Figure 5: We trained CNN on MNIST, MLP on flattened-MNIST, using ADAM (1-cycle, 10 ep), hyperparams determined by random search. Mean (bars: std dev) of $n = 40$ weight inits.

performance across all widths (see Appendix §A.10 for methodology). However when the width is reduced below $2 \times 10^5$ there is a transition and XNOR-shaped activations perform worst. We hypothesize this may be because smaller widths embeddings are over-saturated and have individual units corresponding to multiple features, whilst XNOR activations may require single-feature units to perform best.

With the CNN, five activation configurations ({OR, AND, XNOR$_{\mathrm{AIL}}$ (p)}, {OR, XNOR$_{\mathrm{AIL}}$ (d/p)}, Max, and SiLU) performed best ($p < 0.05$ for other activations, two-tailed Student's $t$-test; Fig. 5, right panel; Appendix §A.10). CNNs which used OR$_{\mathrm{AIL}}$ or Max (alone or in an ensemble) maintained high performance with an order of magnitude fewer parameters ($3 \times 10^4$) than others ($3 \times 10^5$ params).

## 4.4 ResNet50 on CIFAR-10/100

We explored the impact of our activation functions on deep networks by deploying them in a pre-activation ResNet50 model (He et al., 2016a,b). We exchanged all ReLU activations in the network to a candidate activation while maintaining the size of the pass-through embedding. We experimented with changing the width of the network, scaling up the embedding space and all hidden layers by a common factor, $w$. The network was trained on CIFAR-10/-100 for 100 epochs using ADAM (Kingma & Ba, 2015), 1-cycle (Smith, 2018; Smith & Topin, 2017). See Appendix §A.16 for further details.

For both CIFAR-10 and -100, SiLU, OR$_{\mathrm{AIL}}$, and Max outperform ReLU across a wide range of width values (see Fig. 6). These three activation functions hold up their performance best as the number of parameters is reduced. Since SiLU was discovered by a search of $1 \rightarrow 1$ activation functions for this type of architecture and task (Ramachandran et al., 2017), we expected it to perform well in this setting. Yet, we find OR$_{\mathrm{AIL}}$ and Max perform better than ReLU and comparable with SiLU generally. Meanwhile, other AIL activations perform similarly to ReLU when the width is thin, and slightly worse than ReLU when the width is wide. When used on its own and not part of an ensemble, the XNOR$_{\mathrm{AIL}}$ activation function performed poorly (off the bottom of the chart), indicating it is not suited for this task.

## 4.5 Transfer learning

We considered transfer learning on several image classification datasets. We used a ResNet18 model (He et al., 2016a) pretrained on ImageNet-1k. The weights were frozen (not fine-tuned) and used to generate embeddings of samples from other image datasets. We trained a two-layer MLP to classify images from these embeddings using various activation functions. For a compre-

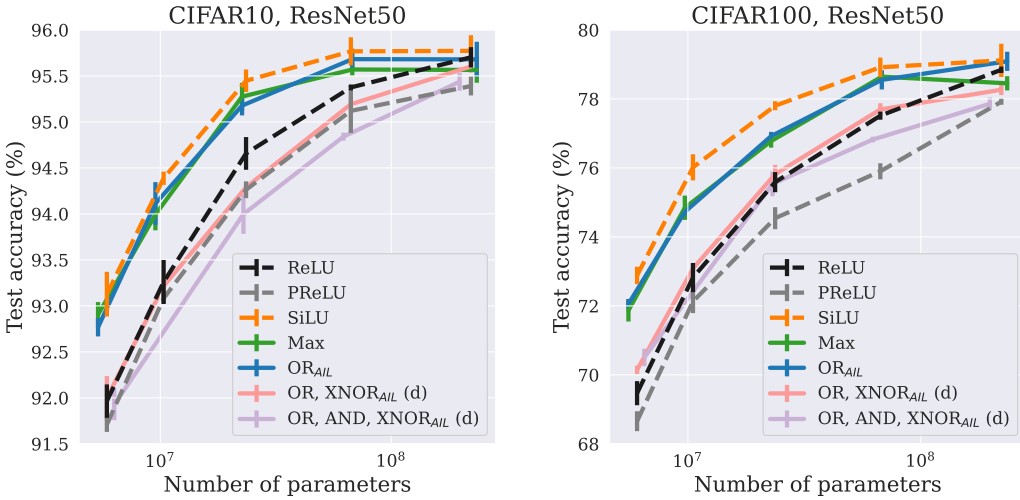

Figure 6: ResNet50 on CIFAR-10/100, varying the activation function used through the network. The width was varied to explore a range of network sizes (see text). Trained for 100 ep. with ADAM, using hyperparams as determined by random search on CIFAR-100 with width factor $w = 2$. Mean (bars: std dev) of $n = 4$ weight inits.

hensive set of baselines, we compared against every activation function built into PyTorch 1.10 (see Appendix §A.17). To make the number of parameters similar, we used a width of 512 for activation functions with $1 \to 1$ mapping (e.g. ReLU), a width of 650 for activation functions with a $2 \to 1$ mapping (e.g. Max, $\mathrm{OR}_{\mathrm{AIL}}$), and a width of 438 for $\{\mathrm{OR}, \mathrm{AND}, \mathrm{XNOR}_{\mathrm{AIL}}$ (d)$\}$. See Appendix §A.17 for more details.

Our results are shown in Table 1. We found that all our (N)AIL activation functions outperformed ReLU on every transfer task. Normalization had little impact on the performance of $\mathrm{OR}_{*\mathrm{IL}}$, but a large impact on $\mathrm{XNOR}_{*\mathrm{IL}}$. The best overall performance was attained by the exact $\mathrm{XNOR}_{\mathrm{NIL}}$, closely followed by our approximate $\mathrm{XNOR}_{\mathrm{NAIL}}$ and ensembles containing either of these. Surprisingly, our approximation $\mathrm{OR}_{*\mathrm{AIL}}$ outperformed the exact form $\mathrm{OR}_{*\mathrm{IL}}$. The proposed activation functions were beaten only by PReLU, on Stanford Cars. On Caltech101, all MLPs were beaten by a linear layer, suggesting our MLPs were overfitting for that dataset. For further discussion, see Appendix §A.17.

### 4.6 Additional results

For results on tabular data, abstract reasoning and compositional zero-shot learning tasks, please see Appendix §A.12, Appendix §A.18 and Appendix §A.19, respectively.

## 5 Conclusion

In this work, we motivated and introduced novel activation functions analogous to Boolean operators in logit-space. We designed the AIL functions, fast approximates to the true logit-space probabilistic Boolean operations, and demonstrated their effectiveness on a wide range of tasks.

Although our activation functions assume independence (which is generally approximately true for pre-activation features learnt with 1D activations), we found the network learnt to induce anti-correlations between features which were paired together by our activations (Appendix §A.14). This suggests exact independence of the features is not essential to the performance of our proposed activations.

We found $\mathrm{XNOR}_{*\mathrm{AIL}}$ was highly effective for shallow MLPs. Meanwhile, $\mathrm{OR}_{\mathrm{AIL}}$ was highly effective for representation learning in the setting of a deep ResNet architecture trained on images. In scenarios which involve manipulating high-level features extracted by an embedding network, we find that using $\mathrm{XNOR}_{*\mathrm{IL}}$ or an ensemble of AIL activation functions together works best, and that the duplication ensembling strategy outperforms partitioning. In this work we restricted ourselves to

Table 1: Transfer learning from a frozen ResNet-18 architecture pretrained on ImageNet-1k to other computer vision datasets. Mean (std. error) of $n = 5$ random initializations of the MLP (same pretrained encoder). Bold: best. Underlined: top two. Italic: no sig. diff. from best (two-sided Student's $t$-test, $p > 0.05$). Background: linear color scale from ReLU baseline (white) to best (black).

| Activation function | Test Accuracy (%) | | | | | | |
|---|---|---|---|---|---|---|---|
| | Cal101 | CIFAR10 | CIFAR100 | Flowers | StfCars | STL-10 | SVHN |
| Linear layer only | $88.35_{\pm0.15}$ | $78.56_{\pm0.09}$ | $57.39_{\pm0.09}$ | $92.32_{\pm0.20}$ | $33.51_{\pm0.06}$ | $94.68_{\pm0.02}$ | $46.60_{\pm0.14}$ |
| ReLU | $86.58_{\pm0.17}$ | $81.63_{\pm0.05}$ | $58.04_{\pm0.11}$ | $90.71_{\pm0.26}$ | $30.97_{\pm0.26}$ | $94.62_{\pm0.06}$ | $53.26_{\pm0.08}$ |
| PReLU | $87.83_{\pm0.21}$ | $81.03_{\pm0.13}$ | $58.90_{\pm0.18}$ | $93.17_{\pm0.19}$ | $39.84_{\pm0.18}$ | $94.54_{\pm0.05}$ | $53.47_{\pm0.08}$ |
| SELU | $87.74_{\pm0.09}$ | $79.93_{\pm0.13}$ | $58.24_{\pm0.06}$ | $92.27_{\pm0.13}$ | $37.51_{\pm0.17}$ | $94.53_{\pm0.07}$ | $50.94_{\pm0.12}$ |
| GELU | $87.10_{\pm0.15}$ | $81.39_{\pm0.09}$ | $58.51_{\pm0.13}$ | $91.51_{\pm0.15}$ | $33.43_{\pm0.15}$ | $94.62_{\pm0.06}$ | $53.43_{\pm0.23}$ |
| Mish | $87.11_{\pm0.12}$ | $81.09_{\pm0.11}$ | $58.37_{\pm0.10}$ | $91.61_{\pm0.15}$ | $33.75_{\pm0.14}$ | $94.61_{\pm0.05}$ | $53.05_{\pm0.12}$ |
| Tanh | $87.48_{\pm0.06}$ | $80.56_{\pm0.07}$ | $57.35_{\pm0.08}$ | $90.32_{\pm0.20}$ | $29.51_{\pm0.12}$ | $94.63_{\pm0.07}$ | $51.86_{\pm0.05}$ |
| Max | $86.96_{\pm0.20}$ | $81.76_{\pm0.14}$ | $58.60_{\pm0.12}$ | $90.98_{\pm0.18}$ | $33.37_{\pm0.15}$ | $94.70_{\pm0.06}$ | $53.53_{\pm0.16}$ |
| Max, Min (d) | $87.23_{\pm0.13}$ | $82.31_{\pm0.10}$ | $59.05_{\pm0.10}$ | $91.68_{\pm0.18}$ | $34.91_{\pm0.12}$ | $94.64_{\pm0.04}$ | $53.91_{\pm0.13}$ |
| SignedGeomean | $87.03_{\pm0.23}$ | $51.45_{\pm16.92}$ | $11.80_{\pm10.80}$ | $91.34_{\pm0.34}$ | $26.37_{\pm6.46}$ | $94.68_{\pm0.06}$ | $37.16_{\pm7.18}$ |
| $\text{XNOR}_{\text{IL}}$ | $85.01_{\pm0.17}$ | $79.62_{\pm0.09}$ | $57.14_{\pm0.07}$ | $84.76_{\pm0.43}$ | $1.34_{\pm0.11}$ | $94.51_{\pm0.03}$ | $51.99_{\pm0.16}$ |
| $\text{OR}_{\text{IL}}$ | $87.11_{\pm0.08}$ | $79.75_{\pm0.05}$ | $58.07_{\pm0.11}$ | $91.12_{\pm0.36}$ | $33.12_{\pm0.12}$ | $94.60_{\pm0.03}$ | $51.21_{\pm0.17}$ |
| $\text{XNOR}_{\text{NIL}}$ | $87.25_{\pm0.22}$ | $82.88_{\pm0.08}$ | $60.78_{\pm0.08}$ | $93.26_{\pm0.26}$ | $39.47_{\pm0.20}$ | $94.83_{\pm0.06}$ | $55.34_{\pm0.19}$ |
| $\text{OR}_{\text{NIL}}$ | $87.19_{\pm0.16}$ | $79.61_{\pm0.05}$ | $58.44_{\pm0.10}$ | $91.65_{\pm0.29}$ | $35.82_{\pm0.04}$ | $94.58_{\pm0.03}$ | $50.95_{\pm0.15}$ |
| $\text{OR, AND}_{\text{NIL}}$ (d) | $86.82_{\pm0.18}$ | $80.09_{\pm0.09}$ | $58.60_{\pm0.07}$ | $91.44_{\pm0.20}$ | $37.03_{\pm0.11}$ | $94.65_{\pm0.05}$ | $52.49_{\pm0.09}$ |
| $\text{OR, XNOR}_{\text{NIL}}$ (d) | $87.82_{\pm0.19}$ | $82.67_{\pm0.05}$ | $60.60_{\pm0.11}$ | $92.93_{\pm0.12}$ | $39.22_{\pm0.17}$ | $94.63_{\pm0.06}$ | $54.87_{\pm0.09}$ |
| $\text{OR, AND, XNOR}_{\text{NIL}}$ (d) | $87.41_{\pm0.27}$ | $82.84_{\pm0.06}$ | $60.38_{\pm0.10}$ | $92.98_{\pm0.17}$ | $39.42_{\pm0.18}$ | $94.71_{\pm0.03}$ | $55.11_{\pm0.12}$ |
| $\text{XNOR}_{\text{AIL}}$ | $86.97_{\pm0.18}$ | $81.83_{\pm0.06}$ | $58.46_{\pm0.10}$ | $90.93_{\pm0.15}$ | $32.56_{\pm0.10}$ | $94.71_{\pm0.06}$ | $53.75_{\pm0.14}$ |
| $\text{OR}_{\text{AIL}}$ | $87.45_{\pm0.14}$ | $81.88_{\pm0.07}$ | $59.10_{\pm0.09}$ | $92.00_{\pm0.15}$ | $36.01_{\pm0.12}$ | $94.69_{\pm0.04}$ | $53.68_{\pm0.14}$ |
| $\text{XNOR}_{\text{NAIL}}$ | $87.61_{\pm0.23}$ | $82.38_{\pm0.07}$ | $59.77_{\pm0.13}$ | $93.07_{\pm0.20}$ | $39.77_{\pm0.04}$ | $94.81_{\pm0.03}$ | $53.91_{\pm0.05}$ |
| $\text{OR}_{\text{NAIL}}$ | $87.19_{\pm0.16}$ | $81.79_{\pm0.09}$ | $59.40_{\pm0.09}$ | $92.12_{\pm0.12}$ | $37.32_{\pm0.17}$ | $94.65_{\pm0.04}$ | $53.82_{\pm0.21}$ |
| $\text{OR, AND}_{\text{NAIL}}$ (d) | $87.62_{\pm0.11}$ | $82.28_{\pm0.10}$ | $59.71_{\pm0.05}$ | $92.10_{\pm0.20}$ | $37.70_{\pm0.12}$ | $94.61_{\pm0.08}$ | $53.86_{\pm0.10}$ |
| $\text{OR, XNOR}_{\text{NAIL}}$ (d) | $87.85_{\pm0.22}$ | $82.52_{\pm0.11}$ | $60.02_{\pm0.10}$ | $93.12_{\pm0.13}$ | $39.64_{\pm0.09}$ | $94.75_{\pm0.03}$ | $54.13_{\pm0.05}$ |
| $\text{OR, AND, XNOR}_{\text{NAIL}}$ (d) | $87.78_{\pm0.14}$ | $82.67_{\pm0.06}$ | $60.01_{\pm0.21}$ | $93.12_{\pm0.21}$ | $39.65_{\pm0.14}$ | $94.78_{\pm0.03}$ | $54.58_{\pm0.12}$ |

only considering using a single activation function (or ensemble) throughout the network, however our results together indicate stronger results may be found by using $\text{OR}_{*\text{AIL}}$ for feature extraction and either $\text{XNOR}_{*\text{IL}}$ or ensemble $\{\text{OR, XNOR}_{*\text{IL}}$ (d)$\}$ for later higher-order reasoning layers.

Our work shows there is more to learn about the importance of more complex activation functions, both for ANN applications and for non-linear dendritic integration in biological neuronal networks.

## Acknowledgments and Disclosure of Funding

We are grateful to Chandramouli Shama Sastry, Eleni Triantafillou, and Finlay Maguire for insightful discussions.

Resources used in preparing this research were provided, in part, by the Province of Ontario, the Government of Canada through CIFAR, and companies sponsoring the Vector Institute https://vectorinstitute.ai/partners/, and in part by ACENET https://ace-net.ca/ and Compute Canada https://www.computecanada.ca/. Additionally, we gratefully acknowledge the support of NVIDIA Corporation with the donation of the Titan Xp GPU used for this research.

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
