# OpenReview forum: "Logical Activation Functions: Logit-space equivalents of Probabilistic Boolean Operators"
_NeurIPS.cc/2022/Conference — NeurIPS 2022 Accept_

### Official Review · Reviewer_BBqK · 2022-07-02

**Rating:** 7
**Confidence:** 4
**Soundness:** 3 good
**Presentation:** 4 excellent
**Contribution:** 3 good

**Summary:**

In this paper the authors presents new activation functions based on a relaxation of the logical operators AND, OR, XNOR. Moreover, the authors present approximations that are more efficient based on simple max, min and addition operations.  This is motivated by the behavior of the operators under the logit space equivalence. As these binary operators are reduction operations, i.e., they convert two inputs into one, the authors connect to the MaxOut. Furthermore, they provide ensemble alternatives to adapt neural network architectures to use the presented boolean activation functions. Finally, the authors provide empirical results showing how the different potential combinations of logical activation functions behave for different datasets and architectures.

**Questions:**

Do you think that these activation functions could provide a path for explainability of neural networks?

I see this paper somehow related to continuous logic functions, and I'm wondering whether composition of more complex logical functions would also make sense, e.g., introducing an implication activation, etc.



**Limitations:**

no negative societal impact

**Strengths And Weaknesses:**

Here, the authors present new activation functions. One weakness could be that they make independency assumptions, however, this does not seem to negatively affect the performance of the models.

The empirical evaluation is sound, and demonstrates the activation functions behave well and even improve the performance in some cases. They achieve this without significantly adding more parameters by using ensembles of activation functions. This is an interesting contribution as previously (MaxOut) would significantly increase the number of weights in order to have the same number of outputs.

---

> ### Author Response · Authors · 2022-08-02
> **Thank you for your feedback**
>
> We thank the reviewer for the favourable review.
>
> We also thank the reviewer for recognizing strengths of this work including the soundness of the evaluation, the effective performance, and the parameter efficiency of the proposed activation functions.
>
> To answer the reviewer’s specific questions:
>
> > * *Do you think that these activation functions could provide a path for explainability of neural networks?*
>
> We do think that this will be useful for increasing the explainability of ANNs, and this is something we would be interested to work on in future. We would like to do so in collaboration with interpretability experts, as none of the present authors have expertise in this area. In our efforts so far, while the behaviour of the activation functions does indeed seem more intelligible, a challenge has been that the embedding space---to which the activation functions are applied---is itself still hard to interpret, so it is still unclear how to better understand what is going on in the middle of a deep network. We believe that coupling these activation functions with existing work in the explainability literature will help bridge this gap, and this is something we would very much like to work on next.
>
> > * *I see this paper somehow related to continuous logic functions, and I'm wondering whether composition of more complex logical functions would also make sense, e.g., introducing an implication activation, etc.*
>
> That's certainly an interesting question! Note that in Boolean logic,
>
> $\mbox{Implication}(p,q) = \mbox{Or}(\mbox{Not}(p),q)$
>
> and our IL and AIL activation functions will have this same property. Since the weight matrix has the flexibility to invert features and perform the logit-space not operation (L178-180), using Implication on its own will yield the same results as using OR on its own (just as we attain the same results for AND as with OR by symmetry). However, it is quite possible that using Implication and OR together in a duplication ensemble is superior to OR and AND, or that Implication and XNOR in an ensemble is better than OR and XNOR, etc. This is something we have not yet investigated, and would be a worthwhile question to explore.

---

### Official Review · Reviewer_VL66 · 2022-07-10

**Rating:** 7
**Confidence:** 2
**Soundness:** 3 good
**Presentation:** 3 good
**Contribution:** 3 good

**Summary:**

This paper presents a new set of activation functions based on the approximation of  logit arithmetic AND/OR/XOR operation. The effectiveness of the proposed activity functions are demonstrated on many tasks including image classification, transfer learning, abstract reasoning, and compositional zero-shot learning.

**Questions:**

* Does it improve or worsen the training time?
* Why is the performance of AIL better than IL in most cases(such as show in Table 1)? Especially in cases like stfCars that XNOR_IL has barely any accuracy.
* Can different layers use different settings? (some use XNOR, some use XNOR/OR d, etc)



**Limitations:**

The authors did discuss the limitations of the proposed method, such as the method being suitable for certain tasks based on experimental results.

**Strengths And Weaknesses:**

**Originality**:

* (+) To my knowledge, the idea proposed in this paper that uses  n->1 activation functions via approximating logit-space boolean operations is novel and interesting

**Quality**:

* (+) The paper technically sounds correct and claims well supported by theoretical analysis and experimental results.
* (+) Related works are covered and discussed.
* (+)  Experiments are conducted extensively in different type of tasks and the results are discussed thoroughly.


**Clarity**:

* (+) This paper is well written and organised.
* (-) it would be better to have a table accompanying figures such as figure 6 as the result can be difficult to read given the amount of legend. Especially for figure 5, 6, the table for legend given is impossible to distinguish between silu and OR, XNOR d



**Significance**:
As demonstrated in the experimental results, the accuracy gain in CIFAR datasets in the transfer learning tests are quite big. I would assume that the proposed activation functions can be quite effective in certain types of tasks.

---

> ### Author Response · Authors · 2022-08-02
> **Thank you for your feedback**
>
> We thank the reviewer for the favourable review.
>
> We also thank the reviewer for recognizing strengths of this work, including the effectiveness of the proposed activation functions, the extent of the evaluation, and the soundness of both the experimental results and the theoretical analysis.
>
> Thank you for the suggestion of adding an accompanying table showing a summary of the results indicated in the plots. We agree this will make the overall results easier to read, and we will be happy to make such a change.
>
> With regards to the reviewer's specific questions:
>
> > * *Does it improve or worsen the training time?*
>
> In terms of wall time to train for X epochs, there is an increase in duration required when using AIL activations versus ReLU, around 10%. However our implementation wasn't very efficient; we've already done some work on improving our pytorch implementation since and it could clearly be much further improved with a cuda implementation, etc.
> In terms of number of epochs to reach a given performance, we saw some initial results showing that $OR_{AIL}$ learnt faster at the start of training than ReLU networks. We hypothesized that this was because the $OR_{AIL}$ gradient is denser than ReLU, as shown in A.5, with the derivative passing through both operands a quarter of the time. But there was no difference in number of epochs to reach peak performance; moreover it was not clear to us that the difference observed was robust to changes in the LR. Hence we didn't explore this effect in detail, nor did we include it in the paper.
>
> > * *Why is the performance of AIL better than IL in most cases(such as show in Table 1)? Especially in cases like stfCars that XNOR_IL has barely any accuracy.*
>
> For XNOR, if no normalization is used then the gain is significantly below 1 because $\sqrt{\mathbb{E}[X^2]}$ is quite small. The problem is worse for XNOR_IL (stdev=0.3664) than for XNOR_AIL (stdev=0.6028). But when we apply normalization, XNOR_IL is elevated from one of the worst-performing actfuns, to XNOR_NIL, the best performing actfun on our transfer learning task. With normalization, the exact form XNOR_NIL performs better than our approximation XNOR_NAIL.
> But for OR_(N)IL vs OR_(N)AIL, the reviewer’s observation is indeed correct - the approximation performs better than the exact form. We were surprised to see this, and currently don't have a hypothesis for why this would be the case.
>
> > * *Can different layers use different settings? (some use XNOR, some use XNOR/OR d, etc)*
>
> Certainly! While we did not explore this in the present paper, we do have preliminary followup work suggesting OR is more useful for early layers and XNOR is more useful for later layers. This is something we are actively working on and very interested in.

---

### Official Review · Reviewer_59zS · 2022-07-11

**Rating:** 6
**Confidence:** 4
**Soundness:** 2 fair
**Presentation:** 2 fair
**Contribution:** 2 fair

**Summary:**

This paper introduces a set of novel activation functions for
artificial neural networks, inspired by the Boolean logic gates for
AND, OR and XNOR.  After defining and motivating the functions, the
paper derives approximated versions that are more computationally
efficient, and demonstrates their use on a few test problems.


**Questions:**

1. Why is it important for network to possess Boolean activation
  functions, given that large enough networks can approximate any
  arbitrary function?
2. Of all the combinations of activation functions explored in the
  experiments section, which one should I pick to use in my next deep
  net architecture? Is there a principle, a strategy or a rule of
  thumb that I could use to choose one (or more) activation function
  without having to train and test my network with all possible
  alternatives?


**Limitations:**

I couldn't find much in terms of explicit analysis of limitations.


**Strengths And Weaknesses:**

### Strengths
The question addressed by the paper is reasonably original,
and the proposed activation functions are derived and described
clearly.

### Weaknesses
In my view the main weakness of the paper is insufficient
motivation and significance. Why is it interesting to have activation
functions that mimic logical gates? No clear motivation is given,
either from a theoretical standpoint (e.g., networks with such
activation functions may be expected to learn more efficiently for
some reason), or from the standpoint of biological inspiration (a few
words are spent on dendritic computation, but logical operations are
not what defines dendritic computation - they are just one particular
abstraction that is sometimes applied to it). At the same time, in the
experiments section there is no clear sign of particular empirical
advantages conferred by the proposed activation functions. Therefore,
to summarize, the logical activation functions, although defined in a
consistent and reasonable way, lack a-priori theoretical motivation as
well as a-posteriori demonstrations of empirical usefulness.

I'll now expand on what I meant by "no sign of empirical advantages"
above. In my view, the results section presents an overly-optimistic
interpretation of the empirical results. Throughout the various
examples, a range of choices of activation functions is explored, but
no choice is shown to be consistently better than the others across
tasks (perhaps with the exception of learning the parity of the
inputs; but that case is more of a sanity check, as it is the natural
operation implemented by the XNOR gate). Indeed:
- in 4.2 (MLP on Bach chorales) there is no statistically significant
  advantage;
- in 4.3 (MNIST) the results are mixed and overinterpreted: for the
  MLP the text says that one of the proposed gates is matched for best
  place with one of the alternatives considered for comparison, but
  this is not even evident from the plot (there are hyperparameter
  regimes where that choice of gate is the one that performs the
  worst). For the CNN case, a different choice of logical-gate
  activations is claimed to be among the ones performing best, but
  again from the plot it looks like the "Max" activation function is
  the best overall.
- in 4.4, the gate that was identified as the best performer in
  4.3/MLP is now the worst performer, so much so that it is not even
  included in the plot. The other gates all produce mixed results with
  no clear winner. The activation that most consistently performs
  better than others in this section seems to be SiLU, but in the text
  it is grouped together with Max and ORAIL (one of the proposed
  gates). For some reason the results of this section don't have error
  bars.
- section 4.5 (transfer learning) also fails to find a scenario where
  the proposed activations consistently perform better than the
  alternatives.

Finally, the experiments section is at times made hard to interpret by
lack of methodological details. For instance, it is not clear to me
how the number of parameters is balanced across activation function
choices, since the different structure of 1-to-1 and 2-to-1
activations (e.g. ReLU vs the activations introduced in this paper)
imply a different organization of the network. Moreover, some p-values
are given without giving any information on the test (such as line
262, "For the MLP, XNORAIL performed best along with signed_geomean (p
< 0.1)", and some results seem to lack any statistical analysis/error
bars (e.g. Figure 6).

---

> ### Author Response · Authors · 2022-08-02
> **Thank you for your feedback.  Overview: Motivation**
>
>
> We thank the reviewer for their feedback and for indicating directions we could emphasize more clearly in a camera-ready version, especially in terms of (I) motivation and (II) empirical results. To make sure we fully understand the nature of the reviewer’s concerns, we hope to open a discussion that will ultimately allow us to address them most effectively in a final version of the paper. We begin with a general reply, and will then provide more detailed discussion, and will use boldface for fast readability.
>
>
>
> ## (I) Re: Motivation:
>
> > *R59zS: “Why is it important for network to possess Boolean activation functions, given that large enough networks can approximate any arbitrary function?”*
>
> As a basic premise, even though sufficiently large networks can indeed approximate arbitrary continuous functions, **we believe there is still good reason to research neural networks and machine learning beyond scaling**. We have assumed that motivating at this level is unnecessary for this venue (NeurIPS); however, if any of the reviewers or AC feel we need to elaborate on this, we are of course happy to discuss in detail and modify the paper accordingly. For now, we build on this premise…
>
> Activation functions are a building block of neural networks, and while they have historical significance (e.g. the switch from sigmoid to ReLU was critical in precipitating subsequent breakthroughs), much remains to be understood about them. **Motivated by both (1) neuroscience and (2) probabilistic reasoning**, we introduce a new class of activation functions which we hypothesize (and demonstrate) to be particularly **appropriate for certain kinds of tasks** (e.g. transfer learning). Furthermore, we hypothesize they will **encourage parameter efficiency in certain circumstances**, and we provide preliminary evidence of this as well. (More details below)

---

> > ### Author Response · Authors · 2022-08-02
> > **Overview: Empirical Results**
> >
> > ## (II) Re: Empirical results:
> >
> > Please note that we conducted **extensive experimentation on an extensive array of tasks (e.g. see Table 3), systematically (multiple seeds, over 15 commonly used activation functions, etc), and we are presenting all of these results**, not only the best tables. We therefore did not expect the entire set of results (i.e. many thousands of runs) to be explainable with one simple story or heuristic. To the best of our knowledge, nobody has conducted a comparison of activation functions as comprehensive as ours on as comprehensive a set of tasks (our comparison alone, although not the focus of our paper, may itself be a valuable contribution to the community). The complexity of the story does not mean that there is a lack of empirical evidence. We compared across all activation functions implemented in PyTorch, and over a dozen variations of the proposed logical activations. Empirical evidence shows that there is no single activation function that is “best” across the board (certainly ReLU is far from best, for example), but that does not mean that activation functions do not matter. On the contrary, activation functions do matter, and as is often the case, the full story appears complicated, with interactions between factors. As can be seen in our results (for example, see Tables 1, 5, 6, 7, 9), our proposed class of activation functions is often better, or among the best-performing, especially on certain families of tasks.
> >
> > > *R59zS:  “Of all the combinations of activation functions explored in the experiments section, which one should I pick to use in my next deep net architecture? Is there a principle, a strategy or a rule of thumb that I could use to choose one (or more) activation function without having to train and test my network with all possible alternatives?” *
> >
> > This is a good question which partially motivated our work in the first place. Our early experiments, before introducing logical actfuns, indicated that no single “common” activation function was best, and the answer was in fact task-dependent; any simple single answer, even among existing/common activation functions, would likely have been wrong. This is shown in our present results. Had we wanted to present a simple view, we could have included fewer experiments, kept the transfer learning experiments—which themselves included 7 tasks x 30 actfuns x 5 seeds x 3 widths = **~3500 runs**—and provided a single take-home message, such as **“Use XNOR for shallow MLPs and Transfer learning”**. Similarly-flavoured statements could be made for other pairs of (activation function, task family) as well, and **we do articulate these observations in our conclusion, in a non-prescriptive way (e.g. see L317–325)**. We stay non-prescriptive precisely because the full story is complicated, and at the time of writing we felt it would be more accurate to summarize our observations as we do in the conclusion. If any of the reviewers or AC feel that the paper would be stronger if we present these observations in the form of rules-of-thumb, we could do so. Overall, we feel this is a very important question that ties back to our motivation in the first place: **there is much to be understood about activation functions** (besides the relationship to, e.g. saturated units, etc), and we are making steps towards this by introducing a probabilistically principled way of constructing them and by extensive and methodical evaluations. We provide more detailed comments on empirical results below.

---

> > > ### Author Response · Authors · 2022-08-02
> > > **Detailed Discussion Points**
> > >
> > >
> > > We next provide additional, more detailed, discussion points.
> > >
> > >
> > >
> > > * **Motivation 1: Neural computation**
> > >
> > > We know from neuroscience research that the current activation functions are at most a very limited approximation of biological neural processes, and it is well established that there are much more complex dendritic computations. It has been shown that biological neurons can compute AND, OR, and even XOR of their inputs. Can we mimic and explore the consequences of this complexity in an artificial neuron's activation function in a way which is scalable? This corresponds to moving some of the complexity of the network from the global structure to the neural level. What are the consequences of this shift? Of course there is far more complexity in biological neurons than the simple abstractions that we consider in this work, but we at least make a step in the direction of using more complex neurons in ANNs. We believe there is value in showing that our systematically constructed and biologically-inspired activation functions are doing as well as other ones while offering a theoretical underpinning. **See L24-31**.
> > >
> > > * **Motivation 2: Probabilistic reasoning**
> > >
> > > We understand neuronal activations in an ANN to correspond to logits (the log odds ratio representation of a probability) representing the presence of a feature within a stimulus. From a Bayesian perspective, a ReLU or ReLU-like operation does not make sense — it is the removal of all evidence for the lack of a feature. Under the logit interpretation of ANN potentiations, this stands out as unreasonable. For example, the truth of this statement can be seen in some of the ways we interact with neural networks — we must apply batch norm before activations and not after them; when doing transfer learning from an embedding space we must use the pre-activation potentiations instead of activations. Can ANNs do better if we design an architecture which treats potentiations as logits? This means our network can perform nested Bayesian reasoning in a similar manner to a Bayesian network, but in logit space using point-estimates of probabilities (and this notion of Bayesian reasoning is very different to a Bayesian neural network). **See L32-36, L45-71**. If we consider (**L851**) there to be two modes of thinking in the brain as described by Kahneman (2011), in which System 1 is instinctual/thinking fast, and System 2 is deliberative/thinking slow, then the connectionist school of ML work corresponds to System 1, and symbolism to System 2. Our paper works towards a way of integrating the two together, and enabling the network to learn symbolic-like reasoning in a connectionist framework.
> > >
> > >
> > > *  **R59zS**: *"[...] networks with such activation functions may be expected to learn more efficiently for some reason [...]"*
> > >
> > > We hypothesize that the logical activation functions are more parameter efficient (**L42, L97**). This is anticipated because (a) we move some of the structure complexity of the network into the activation function; and (b) by not clipping away negative evidence, our actfun outputs are smaller instead of being a vector consisting of zeros for half the values. The choice of activation function corresponds to a structural prior, and we will only be more efficient in situations where that prior holds true despite the fact that our AND and OR activations are generalizations of ReLU to 2d ("no free lunch").
> > >
> > > * **A question regarding Soundness.**
> > >
> > > We note that the reviewer rated our soundness as “poor”. As we made significant efforts to be very systematic, fair, and comprehensive in our evaluations, **we would love to understand better exactly what they found to not be sound**, of course we would be extremely interested and keen to improve that aspect of this paper. (Note that we do explain the missing error bars below, and will fix that).
> > >
> > >
> > > * **R59zs**:  *"For some reason the results of this section don't have error bars."*
> > >
> > > We appreciate that the reviewer noticed this. This was unfortunately due to a crash in which we lost data before we finished finalizing the plots. We can say that there were roughly ~5 seed runs for each, and there were no noticeable outliers, but we are in the process of re-generating this (re-running experiments, etc) to include error bars.
> > >
> > > *  **R59zs**: *"in 4.4, [...] The activation that most consistently performs better than others in this section seems to be SiLU, but in the text it is grouped together with Max and ORAIL (one of the proposed gates)."*
> > >
> > > Thank you for pointing this out. We note that SiLU was discovered by a search of 1d activation functions and optimized for this architecture and task specifically. Given this context, we think it is impressive that OR and Max perform better than ReLU and comparable with SiLU generally, with OR_AIL beating SiLU on CIFAR-10 for wider ResNet architectures. This is no mean feat.

---

> > > > ### Author Response · Authors · 2022-08-02
> > > > **Detailed Discussion Points (cont'd)**
> > > >
> > > >
> > > > * **R59zS**:  *”the experiments section is at times made hard to interpret by lack of methodological details””*
> > > >
> > > > We sincerely regret any difficulty that the reviewer might have had in interpreting the experimental section. As we were unable to fit all the details in the experiments section, we put many details in the Appendices (e.g. normalization in A.7, dataset summaries in A.8, parity experiments in A.10, …, transfer learning in A.16, etc). However, if any reviewer has any recommendations on what details or experiments to remove from the main body in order to make space for specific other details, we would be very happy to consider any such reorganization that might improve the overall clarity of the paper.
> > > >
> > > >
> > > > * **R59zS**: *“in 4.3 (MNIST) the results are mixed and overinterpreted: for the MLP the text says that one of the proposed gates is matched for best place with one of the alternatives considered for comparison,...”*
> > > >
> > > > Thank you for pointing this out. Incidentally, XNOR_AIL is better, but the difference was not statistically significant. More importantly, note that signed_geomean isn't an activation function people actually use, it's another XNOR shaped activation we made up which we included to compare XNOR_AIL against as a harder baseline.
> > > >
> > > > * **R59zS**: *“in 4.3 (MNIST) [...] there are hyperparameter regimes where that choice of gate is the one that performs the worst”*
> > > >
> > > > We appreciate this insightful observation. We do have a hypothesis for this, but we are still testing it. Since we are not yet able to fully substantiate our hypothesis, we felt it would not add value to the paper to present it in a speculative form. However, we would be glad to discuss this, and could potentially add it to the manuscript.
> > > >
> > > >
> > > > * **R59zS**: *"it is not clear to me how the number of parameters is balanced across activation function choices, since the different structure of 1-to-1 and 2-to-1 activations (e.g. ReLU vs the activations introduced in this paper) imply a different organization of the network."*
> > > >
> > > > In order to match the number of parameters, we scale up/down the width of the network(s), keeping the number of layers fixed. **See L258, L277-279, L294-L297, L675, L800-805**. In cases where we needed to do something more complicated than this (because of a need to preserve the size of the embedding space or similar), we mention this explicitly (see **L748-758 and A.18**). For the transfer learning experiments, we also included an additional results table in the appendix (Table 7) showing the performance when we keep the pre-activation width constant and allow the number of parameters to vary instead.
> > > >
> > > > * **R59zS**: *"some p-values are given without giving any information on the test"*
> > > >
> > > > All statistical tests were performed using a two-sided Student’s t-test. We indicated this for some statistical tests (**L244, caption of Table 1, etc**), but for brevity did not repeat this description for the three statistical tests appearing in **L262-267**. We apologize for being too liberal with the removal of this detail, and would be glad to reinsert a mention of the test used in this paragraph.
> > > >
> > > >
> > > > Thank you again to the reviewer for providing all of this feedback.

---

> > > > > ### Comment · Reviewer_59zS · 2022-08-06
> > > > > **Thanks for the response**
> > > > >
> > > > > Thanks for your response to my comments.
> > > > >
> > > > > ### Summary
> > > > > Thanks for helping me understand better the motivation behind
> > > > > this study - this partially addresses one of my concerns, in a way
> > > > > that positively affects my evaluation of the remaining outstanding
> > > > > points. I still have issues with how some of the results are
> > > > > interpreted, and with the biological inspiration angle, but this is
> > > > > now less of a problem in my view. I'll update my score accordingly.

---

> > > > > > ### Comment · Reviewer_59zS · 2022-08-06
> > > > > > **Thanks for the response - part 2**
> > > > > >
> > > > > >
> > > > > > ### Motivation
> > > > > >
> > > > > > After reading the response, I understand better the motivation angle
> > > > > > discussed under the title "neural computation". While I don't
> > > > > > necessarily see the link with Kahnemann, I now appreciate better this
> > > > > > possible theoretical motivation for the present study. This addresses
> > > > > > one of my main concerns. It would be great if you could find a way of
> > > > > > emphasising this motivation even more in the text. I feel that it
> > > > > > would improve the impact of the paper, as the stronger the theoretical
> > > > > > motivations are, the less likely it will be that the reader will be
> > > > > > disappointed or confused to see that there is not a clear results
> > > > > > story or simple recommendations or rules of thumb to be gained.
> > > > > >
> > > > > > For the record, I still don't agree with the idea that logical
> > > > > > activation functions move us towards more biologically-inspired
> > > > > > architectures. As I stated in my initial review, performing boolean
> > > > > > operations is something that certain neurons have been shown to be
> > > > > > able to do, but this is far from a generally-useful abstraction of
> > > > > > dendritic computation - at least not a better one than ReLU for
> > > > > > neurons in the rate coding regime. But this is not a big deal - I
> > > > > > appreciate that this can be read as a partial additional motivation
> > > > > > that does not need to stand on its own.
> > > > > >
> > > > > > ### Results
> > > > > >
> > > > > > I am sorry if my soundness rating was confusing. In my review I stated
> > > > > > several passages where I believed the performance of the proposed
> > > > > > gates was presented in an overly optimistic light, which was not
> > > > > > supported by the data. I also expressed concerns about missing
> > > > > > statistics (the error bars in Figure 6) and unclear hypothesis testing
> > > > > > procedures. These were the reasons for the score. I am surprised by
> > > > > > your confusion: in your rebuttal you spend several paragraphs
> > > > > > countering these methodological and interpretational points,
> > > > > > immediately after stating that you would love to know what I have not
> > > > > > found to be sound. However, I thank you for partially addressing some
> > > > > > of them in your response (in particular, point taken about SiLU -
> > > > > > perhaps you could make that a bit more explicit in the text). Some,
> > > > > > however, are still outstanding:
> > > > > >
> > > > > > - you did not address my objection to line 243, where it is stated
> > > > > >   that *"{OR, AND, XNORAIL} performed best"*, while there is no
> > > > > >   statistical evidence to back this up.
> > > > > > - your response to my objection that *"in 4.3 (MNIST) the results are
> > > > > >   mixed and overinterpreted: for the MLP the text says that one of the
> > > > > >   proposed gates is matched for best place with one of the
> > > > > >   alternatives considered for comparison, but this is not even evident
> > > > > >   from the plot (there are hyperparameter regimes where that choice of
> > > > > >   gate is the one that performs the worst)"* addresses the premise, and
> > > > > >   not the actual content, of my criticism. I don't particularly care
> > > > > >   what the alternative being matched for first place is - I care that
> > > > > >   **the general trend of XNORAIL and signed geomean being better is not
> > > > > >   supported by the data in the plot**. Indeed, as you agreed in your
> > > > > >   rebuttal, the supposedly "best" gates are those that perform the
> > > > > >   worst in certain parameter regimes. You say that you have an
> > > > > >   hypothesis for this, but again in this context I don't particularly
> > > > > >   care what the reason may be - I care that XNORAIL is the worst
> > > > > >   performing gate in some regimes, but is described as *"best
> > > > > >   performing"* in the text. You give a p-value for this statement, but
> > > > > >   this is not very meaningful given the missing information about the
> > > > > >   test (which you have now amended, and I will comment on below).
> > > > > > - you did not address my criticism that *"For the CNN case, a different
> > > > > >   choice of logical-gate activations is claimed to be among the ones
> > > > > >   performing best, but again from the plot it looks like the "Max"
> > > > > >   activation function is the best overall."*
> > > > > > - thank you for clarifying that the p-values were computed with a
> > > > > >   two-tailed t-test, but I don't get how this particular test could be
> > > > > >   relevant or applicable to the statement to which the p-value is
> > > > > >   associated. For concreteness, let's take the statement I was
> > > > > >   criticizing above. This is at line 262-263: *"For the MLP, XNORAIL
> > > > > >   performed best along with signed_geomean (p < 0.1), ahead of all
> > > > > >   other activations (p < 0.01)."* How can the second p-value be the
> > > > > >   result of a t-test? How is the t-statistic computed here? This would
> > > > > >   be easy to do (albeit perhaps in a debatable way) if one was
> > > > > >   comparing just two gates: consider the paired differences in
> > > > > >   performance between the two gates, and perform the test on that. But
> > > > > >   how do you use a t-test to compare those two gates to all others?
> > > > > >   This is not immediately evident to me.

---

> > > > > > > ### Comment · Reviewer_59zS · 2022-08-06
> > > > > > > **Thanks for the response - part 3**
> > > > > > >
> > > > > > > ### Other points:
> > > > > > >
> > > > > > > Regarding my question *"Why is it important for network to possess
> > > > > > > Boolean activation functions, given that large enough networks can
> > > > > > > approximate any arbitrary function?"*, to which you replied with a
> > > > > > > series of remarks that can be summarized with *"there is still good
> > > > > > > reason to research neural networks and machine learning beyond
> > > > > > > scaling"*. My point was **"why boolean functions in particular"**, not **"why
> > > > > > > do any research in ML or ANNs at all beyond scaling"**. I apologise if
> > > > > > > something in my wording suggested the latter.

---

> > > > > > > > ### Author Response · Authors · 2022-08-09
> > > > > > > > **Thank you again for your feedback**
> > > > > > > >
> > > > > > > >
> > > > > > > > # Thank you and Overview
> > > > > > > >
> > > > > > > > We would first like to thank R59zS very much for responding both positively and in so much detail to our rebuttal. Their attentiveness and thoughtfulness is appreciated, and helps make the entire review process a positive experience.
> > > > > > > >
> > > > > > > > We will respond below to the new comments, and we are also about to upload a revised manuscript that integrates changes based on this discussion while staying within the page limits. In the revised manuscript, we have indicated new text in blue where possible. We will of course be glad to continue modifying it to incorporate anything not yet addressed.
> > > > > > > >
> > > > > > > > # Detailed Responses
> > > > > > > >
> > > > > > > > > *It would be great if you could find a way of emphasising this motivation even more in the text. I feel that it would improve the impact of the paper, as the stronger the theoretical motivations are, the less likely it will be that the reader will be disappointed or confused to see that there is not a clear results story or simple recommendations or rules of thumb to be gained.*
> > > > > > > >
> > > > > > > > Yes, we have done a first draft of this in our revision, thank you.
> > > > > > > >
> > > > > > > > > *For the record, I still don't agree with the idea that logical activation functions move us towards more biologically-inspired architectures. As I stated in my initial review, performing boolean operations is something that certain neurons have been shown to be able to do, but this is far from a generally-useful abstraction of dendritic computation - at least not a better one than ReLU for neurons in the rate coding regime. But this is not a big deal - I appreciate that this can be read as a partial additional motivation that does not need to stand on its own.*
> > > > > > > >
> > > > > > > > We appreciate the way that R59zS has articulated this fair perspective.
> > > > > > > >
> > > > > > > > > *However, I thank you for partially addressing some of them in your response (in particular, point taken about SiLU - perhaps you could make that a bit more explicit in the text).*
> > > > > > > >
> > > > > > > > We have modified the text accordingly.
> > > > > > > >
> > > > > > > > > *you did not address my objection to line 243, where it is stated that "{OR, AND, XNORAIL} performed best", while there is no statistical evidence to back this up.*
> > > > > > > >
> > > > > > > > Here, we were trying to say that these ones did have the highest accuracy, but that the difference was not statistically significant. Hence the second half of that sentence read "but [..] overall the results were comparable (p < 0.1 between best and worst, Student’s t-test, 10 random inits)", which we admit was muddy and not making it clear what we mean to say.
> > > > > > > >
> > > > > > > > We have changed the wording here to clarify this. Thank you for pointing this out.
> > > > > > > >
> > > > > > > >
> > > > > > > > (cont'd in next part)

---

> > > > > > > > > ### Author Response · Authors · 2022-08-09
> > > > > > > > > **Detailed response (continued 1)**
> > > > > > > > >
> > > > > > > > > **New appendix**. For the next three comments, we are responding in two ways:
> > > > > > > > >
> > > > > > > > > (1) for clarity of the rebuttal process, we are responding directly below for each point; and also,
> > > > > > > > >
> > > > > > > > > (2) In the revision we are uploading, we have added a new appendix that provides the relevant information.
> > > > > > > > > We are splitting new details between the main body and the appendix to balance page limits.
> > > > > > > > > If R59zS feels that we ought to put certain details into the main body, we will be glad to modify further.
> > > > > > > > >
> > > > > > > > > > *thank you for clarifying that the p-values were computed with a two-tailed t-test, but I don't get how this particular test could be relevant or applicable to the statement to which the p-value is associated. For concreteness, let's take the statement I was criticizing above. This is at line 262-263: "For the MLP, XNORAIL performed best along with signed_geomean (p < 0.1), ahead of all other activations (p < 0.01)." How can the second p-value be the result of a t-test? How is the t-statistic computed here? This would be easy to do (albeit perhaps in a debatable way) if one was comparing just two gates: consider the paired differences in performance between the two gates, and perform the test on that. But how do you use a t-test to compare those two gates to all others? This is not immediately evident to me.*
> > > > > > > > >
> > > > > > > > > Thank you for clarifying the question for us. The procedure used was as follows. First we collapsed across the num. params dimension by selecting the best width value (i.e. number of params) for each activation function. Typically this is the widest value, but sometimes some actfuns had their performance peak with fewer parameters (especially on Covertype, where we disabled weight decay). Then we identified the best performing activation function as the one with the highest accuracy. In the case of MLP on MNIST, this was XNOR_AIL. We compared its performance to each of the other activation functions using a (non-paired) Student's t-test, implemented with scipy.stats.ttest_ind, one test for each of the other activations. The total number of tests performed is (num_actfuns - 1). We did not use a Bonferroni correction for multiple comparisons. In the case of the MLP on MNIST, the p-value vs signed_geomean was between 0.05 and 0.1 and hence did not meet our threshold for statistical significance. For all the other activations, the p-value was less than 0.01, more than exceeding our threshold for significance. Hence we are trying to say that XNOR_AIL was significantly better than all other activations besides signed_geomean. We did not perform a t-test between signed_geomean and the other activation functions, and we were not trying to make a claim about the performance of signed_geomean being significantly different from any of the other activation functions.
> > > > > > > > >
> > > > > > > > > > *your response to my objection that "in 4.3 (MNIST) the results are mixed and overinterpreted: for the MLP the text says that one of the proposed gates is matched for best place with one of the alternatives considered for comparison, but this is not even evident from the plot (there are hyperparameter regimes where that choice of gate is the one that performs the worst)" addresses the premise, and not the actual content, of my criticism. I don't particularly care what the alternative being matched for first place is - I care that the general trend of XNORAIL and signed geomean being better is not supported by the data in the plot. Indeed, as you agreed in your rebuttal, the supposedly "best" gates are those that perform the worst in certain parameter regimes. You say that you have an hypothesis for this, but again in this context I don't particularly care what the reason may be - I care that XNORAIL is the worst performing gate in some regimes, but is described as "best performing" in the text. You give a p-value for this statement, but this is not very meaningful given the missing information about the test (which you have now amended, and I will comment on below).*
> > > > > > > > >
> > > > > > > > > We clarified the methodology for the statistical tests in the comment above. Now with regards to the reasoning behind this methodology (collapsing across the num. params by selecting the best for each actfun): it compares each of the actfuns at their best. This is in keeping with how results are frequently reported in ML literature (in a table showing only one value per comparator, after performing a hyperparam search). Simplifying the results by collapsing the num params dimension makes it possible to do a simple statistical test, such as a t-test. But we agree that such a simplification does not tell the whole story about the results. We have amended the discussion of these results to make it clear that our statistical tests are comparing each of the actfuns in their "best-case" scenario, and to discuss the fact that XNOR shaped actfuns lose performance sooner than other actfuns when the num params is reduced.

---

> > > > > > > > > > ### Author Response · Authors · 2022-08-09
> > > > > > > > > > **Detailed response (continued 2)**
> > > > > > > > > >
> > > > > > > > > > > *you did not address my criticism that "For the CNN case, a different choice of logical-gate activations is claimed to be among the ones performing best, but again from the plot it looks like the "Max" activation function is the best overall."*
> > > > > > > > > >
> > > > > > > > > > The methodology here was the same as described above for the MLP. Selecting the best width for each actfun, we find there are 5 actfuns/ensembles which are indistinguishably best performance, Max among them, and others had significantly lower accuracy than the best. L263-265: "With the CNN, five activation configurations ({OR, AND, XNOR AIL (p)}, {OR, XNOR AIL (d/p)}, Max, and SiLU) performed best (p < 0.05; Figure 5, right panel)." So in the text, we are already claiming that Max is one of the best actfuns on the task. However, the outcome which we find more interesting is that some actfuns maintain high performance even as the number of parameters is reduced, and others do not. Hence we then go on to say that "Additionally, we note that CNNs which used OR AIL or Max (alone or in an ensemble) maintained high performance with an order of magnitude fewer parameters (3 × 10^4 ) than networks which did not (3 × 10^5 params)." R-59zS is correct that the line for the Max actfun sits above all others in the 10^4 to 3×10^5 params regime by a small but consistent margin. We have not performed an evaluation of this difference, and therefore cannot comment on it in more detail. To us, the more interesting result is the dichotomy between actfuns which include Max/OR operations compared to those which do not. (The only time the effect size is substantive is at ~1.1x10^5 params, but at that point the Max line is an interpolation and not a measurement - all actfuns measured with this num params had a dip in accuracy, the cause of which we were unable to identify but we suspect it could be a side effect of the MLP configuration changing in relation to L755-L757.)
> > > > > > > > > >
> > > > > > > > > > We have integrated the above three comments and replies into the revised manuscript.
> > > > > > > > > >
> > > > > > > > > > > *Regarding my question "Why is it important for network to possess Boolean activation functions, given that large enough networks can approximate any arbitrary function?", to which you replied with a series of remarks that can be summarized with "there is still good reason to research neural networks and machine learning beyond scaling". My point was "why boolean functions in particular", not "why do any research in ML or ANNs at all beyond scaling". I apologise if something in my wording suggested the latter.*
> > > > > > > > > >
> > > > > > > > > > Thank you very much for this clarification. This makes sense and is consistent with the thoughtful and helpful comments and questions in R59zS’s review. We believe we have addressed this throughout our detailed replies.
> > > > > > > > > >
> > > > > > > > > >
> > > > > > > > > > ### Summary
> > > > > > > > > >
> > > > > > > > > > Thank you again for this positive and helpful review and for engaging (and thus providing us an opportunity to engage as well) in this discussion. We appreciate it.

---

> > > > > > > > > > > ### Comment · Reviewer_59zS · 2022-08-09
> > > > > > > > > > > **Response to authors**
> > > > > > > > > > >
> > > > > > > > > > > Thank you for your response, and for considering my feedback in the revised version of the manuscript. My original concerns have now largely been addressed, and to reflect that I will update my score from 3 to 6.

---

### Meta-Review · Area_Chair_FJfV · 2022-08-30

**Recommendation:** Accept
**Confidence:** Certain

**Metareview:**

The review ratings were above the acceptance threshold. The reviewers valued quite positively the originality of this paper that studied advantages of multivariate activate functions, as well as extensive numerical experiments across a wide range of tasks in order to explore their effectiveness. Upon reading the reviews, the author responses, subsequent discussion between the reviewers and the authors, as well as the paper itself, I thought that the restriction of the consideration in this paper to those derived from approximation of Boolean operators on independent Bernoullis was not well motivated nor described. At the same time, the empirical evidences demonstrating the potential usefulness of the proposal are quite interesting, so that I would like to recommend acceptance of this paper, and would expect further discussion on this subject among the attendees of the conference.

**Award:**

No

---

### Decision · Program_Chairs · 2022-09-14

Accept